# Preparation of Bio-Foam Material from Steam-Exploded Corn Straw by In Situ Esterification Modification

**DOI:** 10.3390/polym15092222

**Published:** 2023-05-08

**Authors:** Yu Pan, Yufan Zhou, Xiaoqing Du, Wangjie Xu, Yuan Lu, Feng Wang, Man Jiang

**Affiliations:** 1Key Laboratory of Advanced Technologies of Materials (Ministry of Education), Chengdu 610031, China; 2School of Materials Science and Engineering, Southwest Jiaotong University, Chengdu 610031, China; 3School of Chemistry, Southwest Jiaotong University, Chengdu 610031, China

**Keywords:** bio-foam material, corn straw, in situ esterification, heat insulation, flame retardant property

## Abstract

In this work, we engineered a corn-straw-based bio-foam material under the inspiration of the intrinsic morphology of the corn stem. The explosion pretreatment was applied to obtain a fibrillated cellulose starting material rich in lignin. The in situ esterification of cellulose was adopted to improve the cross-linking network of the as-developed foam bio-material. The esterification of lignin was observed in the same procedure, which provides a better cross-linking interaction. The esterified corn-straw-derived bio-foam material showed excellent elastic resilience performance with an elastic recovery ratio of 83% and an elastic modulus of 20 kPa. Meanwhile, with surface modification by hexachlorocyclotriphosphazene-functionalized lignin as the flame retardant (Lig-HCCP), the as-obtained bio-foam material demonstrated quite a good flame retardancy (with 27.3% of the LOI), as well as a heat insulation property. The corn-straw-derived bio-foam material is prospected to be a potential substitution packaging material for widely used petroleum-derived products. This work provides a new value-added application of the abundant agricultural straw biomass resources.

## 1. Introduction

Porous polymer materials have applications in extensive aspects, such as purification, separation, drug or fertilizer delivery, and packaging material, etc. In consideration of only packaging materials, there is a huge demand for lightweight plastics, such as polyethylene (PE) and polypropylene (PP), as well as synthetic polymer foams, such as polyurethane (PU) and polystyrene (PS), at present and a continuous increasing trend in the near future with changes in lifestyle and the increasing transportation of commodities. It is estimated that about 265 million tons of synthetic plastic are produced each year, while only 0.7 million tons of biodegradable plastic are made annually [1]. As a result, the accumulation of plastic disposal in the environment pollutes the ecosystems of both terrestrial and ocean environments globally, which has been recognized as the most serious environmental pollution faced by mankind in this century. Both researchers and industry are devoted to implementing sustainable substitutions for petroleum-derived synthetic plastics. Biodegradable plastics are considered as the promising solution, which has been intensively studied, and some commercial products have been realized [2,3,4,5]. Among different types of degradable plastics, starch-based biodegradable plastics are the most widely developed [6,7], which are mainly divided into two categories. The first category is the degradation of starch into monomers, which are then converted into polylacticide (PLA) or polygloycolic (PGA) plastics; the other category is the introduction of starch as the main component to prepare composite plastics. The latter one plays the dominant role currently. Starch is commonly introduced into the polyvinyl alcohol (PVA) matrix, which has the properties of high elongation at break and light transmittance [8,9]. To improve the processability of the composites, different types of esterified starch are applied, or bio-sourced proteins are introduced, such as gliadin and keratin, as additives at the same time [10,11,12]. The application of non-edible natural resources to develop biocompatible and biodegradable materials represents a better way to avoid the potential risks of competition with food [13,14]. Cellulose is the most abundant biopolymer on Earth, which is interesting for its mechanical properties and thermal stability. Researchers have adopted isolated cellulose from biomass resources as reinforcement in thermoplastic corn starch films [15,16], and starch/PVA bio-composites [17,18]. Cellulose-based porous materials have also received significant attention because of their ease of tuning the pore size and functionalization, which are beneficial for meeting the requirement of a variety of practical applications [19,20,21]. A holistic lignocellulosic-biomass-derived biodegradable material from rice straw has been successfully prepared based on the reinforcement of crystalline cellulose microfibrils and the film-forming ability of the hemicelluloses and lignin mixtures [22], which boosted our interest to fabricate a 3D foam material completely from agricultural residues.

Considering the preparation techniques for porous organic polymer scaffolds, supercritical carbon dioxide foaming [23,24], hard template, and soft template methods [25] are basically selected as needed. However, all of them have some limitations for the fabrication of biomass-based porous materials due to their heterogeneity, low reactivity, and low solubility. A facile ice template method has been reported to successfully prepare porous starch with tunable anisotropic morphology, mechanical strength, and low thermal conductivity by using waxy maize starch as the only starting material. By tuning the concentration of starch, which mainly consists of amylopectin, interconnected starch lamellas are formed after unidirectional freezing followed by ice sublimation [26]. Lavoine and coworkers reported a new dual-templating approach using ice and cellulose nanofibrils (CNFs) as templates to control the porous architecture of the biomass-based foams where the CNFs considerably enhanced the structural and mechanical integrity [27]. In most cases, a cross-linking agent is introduced to obtain a stable network by forming covalent bonds or ionic linkages. Moreover, a functional cross linker has been developed to adjust the hydrophilic and hydrophobic properties of the porous materials and to realize exquisite control over the pores [28,29]. As one of the main components of lignocellulosic biomass, lignin typically constitutes 20–32% by weight [30]. The unique polyphenol structure of lignocellulosic biomass renders lignin with many functional properties, such as ultraviolet blocking, antioxidant, antibacterial, as well as flame retardant. With the intention of packaging application safety, flame retardance is considered in this work.

Lignin has considerable potential in the development of sustainable flame retardants for polymers due to its aromatic skeleton and high char-forming capability after combustion [31,32,33]. However, bio-based materials have inherent downsides, partly because of their increased flammability when subjected to heat flux or flame initiators, which limits their range of application [34]. Therefore, we aimed to identify an efficient method to apply the flame retardance of lignin to improve the comprehensive performance of bio-composites. The modification of lignin with functional groups containing N and P [35] and using lignin at nanoscale (LNP) to prepare a functionalized new flame retardant has been previously reported [36].

The present work investigates the feasible engineering of agricultural straw into a foam material as a potential sustainable alternative to non-biodegradable petroleum-based products, for example, packaging applications for shipping convenience and safety. Under the inspiration of the ordered porous architecture of the inner part of the corn stem, which presents a certain compression and flexural resilience, we adopt corn straw as the raw material to prepare a bio-foam material using a dissolution–regeneration method together with freeze drying. Lignin was modified with HCCP and introduced into the as-prepared bio-foam material as a flame retardant.

## 2. Materials and Methods

### 2.1. Materials

Cotton pulp cellulose was purchased from Xinxiang Chemical Fiber Co., Ltd. (Xinxiang, China), and dealkaline lignin and chloroform were purchased from Shanghai Macklin Biochemical Co., Ltd. The vinyl acetate (VA), dimethyl sulfoxide (DMSO), dimethyl formamide (DMF), carboxymethylcellulose sodium (CMCNa), calcium disodium edetate (EDTACa), hexachlorocyclotriphosphazene (HCCP), and sodium hydroxide (NaOH) were purchased from Chengdu Kelong Chemical Reagent Factory. The tetrabutylammonium hydroxide aqueous solution (TBAH, 40 wt%) was obtained from Zhenjiang Runjing High Purity Chemical Co., Ltd. (Zhenjiang, China), which was condensed to 50 wt% with a rotary evaporator (40 °C) before use.

The corn straw was collected from the suburb of Chengdu, Sichuan. The corn straw was detected to contain 49.7 wt% cellulose, 29.8 wt% hemicelluloses, 11.6 wt% lignin, 7.1 wt% lipids, and 1.8 wt% ash. It was steam exploded with a self-made machine under the conditions of 2.2 MPa and 30 min of steaming time. After boiling and washing with deionized (DI) water, the product was dried to obtain the steam-exploded corn straw (SEC). The SEC was tested to contain 63.5 wt% cellulose, 2.3 wt% hemicelluloses, 15.6 wt% lignin, 13.9 wt% lipids, and 4.7 wt% ash. A cellulose analyzer (CA, FIWE Advance, Italy) was used to analyze the main components of the raw corn straw and SEC according to the Van Soest method.

### 2.2. Esterification of Steam-Exploded Corn Straw

As included in the Appendix A, the conditions for the esterification of cellulose with VA as the esterification reagent was preoptimized before conducting the in situ esterification of the corn straw. Firstly, 1 g of SEC was dispersed in 40 mL of DMSO at 60 °C and stirred for 15 min. Then, 2 mL of 10 mol/L NaOH solution was added and stirred for 5 min. At the same time, a certain amount of VA was added to continue the reaction under 120 °C for 20 min, and then ethanol was added at the end of the reaction. It was filtrated and the solid content was washed with ethanol and water separately several times. After being dried in ovens under 60 °C for 24 h, the esterified corn straw (ECS) samples were obtained. Concretely, a three-factor and four-level orthogonal experiment was designed and conducted as shown in Appendix A, and the detailed conditions and results are shown in Appendix A. The pretreatment conditions of temperature, time, and the amount of esterification reagent (VA) on the reaction efficiency were all studied. The samples were named ECS_−x_ according to the esterification efficiency (x: the number represents the degree of substitution).

### 2.3. Preparation of the Bio-Foams from Steam-Exploded Corn Straw

The preparation of the bio-foams from steam-exploded corn straw was according to a common procedure as follows: 2 g of TBAH (50 wt%), 8 g of DMSO, and 0.2 g of SEC or ECS_−x_ were added into a reaction bottle and mixed under magnetic stirring at 60 °C. The cross-linking agents, which consist of 0.0216 g of CMCNa and 0.0144 g of EDTA in 1.2 g of water, were then added. The mixture was poured into a cylinder mold of 1.5 cm × 2 cm (diameter × height). After standing overnight, solvent exchange was conducted to obtain a wet gel using ethanol. Finally, the samples were dried by a freeze dryer at −50 °C under the pressure of 18 Pa for 48 h, and the bio-foams were obtained. Meanwhile, the bio-foams prepared from SEC and ECS_−x_ were respectively named as SEC foam and ECS_−x_ foam.

### 2.4. Flame Retardant Modification of the Corn Straw Bio-Foam

The flame retardant was synthesized as detailed in the following procedure. Dealkali lignin (0.5 g) was dispersed in DMF (15 mL) to be fully dissolved and then pyridine (2 g) was added. By adding a solution of HCCP (1.5 g) in 5 mL DMF, the mixture was allowed to stir at 80 °C for 8 h. Finally, the product was separated by centrifugation at 8000 rpm for 10 min, followed by washing with alcohol and DI water. The purified product was obtained after freeze drying at −50 °C under the pressure of 18 Pa for 48 h, and the resultant product was named Lig-HCCP.

The preparation of the flame-retardant corn-straw-derived bio-foam material was conducted in a common process as follows. After preparing 2 wt% ECS_−4.5_ solution (the solvent was TBAH: DMSO with a mass ratio of 2:8), the cross-linking agents EDTA and CMC were added. Then, ECS_−4.5_ was fully dissolved in the flame retardant Lig-HCCP (the mass ratio of Lig-HCCP to straw acetate was 2:8). Finally, the resulting mixture was poured into a mold, regenerated, dialyzed, and then freeze dried to obtain the bio-foam, which was labeled the Lig-HCCP-ECS_−4.5_ foam.

### 2.5. Structure and Morphology Analysis

A Fourier-transform infrared spectrometer (FTIR, Bruker Tensor Ⅱ, Germany) was used to analyze the chemical structure of the specimens under various conditions. FTIR spectra were collected at a spectral resolution of 4 cm^–1^, with 32 scans over the range from 4000 to 400 cm^–1^.

The aggregate structures of the samples were detected on an X-ray diffractometer (X ‘Pert Pro, Panaco, Almelo, The Netherlands). The test adopted a copper target, and the receiving slit was 0.3 mm, the operating voltage was 40 kV, the current was 30 mA, the scanning speed was 10°/min, and the scanning range was set from 5 to 60°. The crystallinity index (CrI) was calculated by Segal’s equation [37]:(1)$$\mathrm{CrI}=\frac{I_{002}-I_{\mathrm{am}}}{I_{002}}\times 100\%$$
where I_002_ is the maximum intensity at 22.5°, and I_am_ is the lowest intensity of the amorphous area at 18.7°.

Solid-state cross-polarization/magic-angle spinning (CP/MAS) ^13^C NMR spectra of esterified cellulose and lignin were collected with a Bruker Advance III 400 MHz spectrometer at room temperature with 1940 scans and a receiver gain of 1150 (the result is shown in Appendix A).

Scanning electron microscopy (SEM, COXEM600, Daejeon, Korea) was used to observe the microstructure of the materials. The samples were brittle broken and fixed on the copper sample stage. After spraying gold with the ion sputtering instrument (SBC-12, China Science & Technology Instrument, Beijing, China), the samples were detected under a voltage of 12 kV.

### 2.6. Thermal Performance Analysis

The thermal properties of the samples were performed by DSC (Juper STA 449C, NETZSCH, Selb, Germany). The temperature ranged from −10 °C to 280 °C and the heating rate was 10 °C/min in a nitrogen atmosphere. The thermal stability was also tested by a thermal gravimetric analyzer (STA 449 C, NETZSCH, Selb, Germany) with a heating rate of 10 °C/min; the temperature ranged from 0 °C to 800 °C in a nitrogen atmosphere.

### 2.7. Compressive Mechanical Analysis

The compressive properties of the bio-foams were tested by an electronic universal testing machine (CMT4000, Sansi-Taijie, Zhuhai, China) at room temperature at a rate of 5 mm/min.

### 2.8. The Degree of Substitution (DS) Analysis

A chemical reaction method was adopted to test the esterification efficiency, which was the degree of substitution of -OH. Amounts of 0.2 g of ECS and 10 mL of aqueous ethanol (70 vt%) were mixed and stirred for 30 min. Then, 5 mL of NaOH solution (0.5 N) was also added and the mixture was stirred for 48 h at 60 °C. At last, the superfluous NaOH was neutralized with 0.1 N HCl:(2)$$\mathrm{DS}=\frac{\left(5\times 0.5-0.1V_{e}\right)-\left(5\times 0.5-0.1V_{c}\right)}{0.2-\left[\left(5\times 0.5-0.1V_{e}\right)-\left(5\times 0.5-0.1V_{c}\right)\right]\times m\times 10^{-3}}\;\left({\mathrm{mmol}\;g}^{-1}\right)$$
where V_e_ and V_c_ respectively represent the volumes (mL) of HCl solution added to the esterified straw and straw (control). The m indicates the molar mass of the attached acetyl residue [38].

### 2.9. The Simulation Analysis Method with Finite Element Procedures

The thermal conductivity of the bio-foams was tested by a thermal constant analyzer (TPS2200, HotDisk, Goteborg, Sweden) at room temperature. The molar heat capacity at constant pressure was tested by DSC (Q600, TA instrument, Newcastle, TL, USA). The true density of the bio-foams was measured by ULTRAPYC1200e (Kanta Instrument, Florida, USA) to calculate the porosity using the method of inert gas replacement (helium) at 100 °C. COMSOL software was used to simulate the thermal insulation performance of Lig-HCCP-SECs_−4.5_ foam at 60 °C and 100 °C. The simulation parameters were obtained according to the above tests. The parameters were shown in the following Appendix A.

### 2.10. Characterization of Flame Retardancy

The limiting oxygen index (LOI) is the lowest volume fraction concentration of oxygen in a mixture of oxygen and nitrogen when the polymers start to burn at room temperature. It was measured at room temperature by a TTech-GBT2406-1 intelligent critical oxygen index analyzer according to GB/T 2406.2-2009 standard with specimen dimensions of 100 mm × 10 mm × 10 mm.

## 3. Results and Discussion

### 3.1. Characterization of Esterified Steam-Exploded Corn Straw

#### 3.1.1. Morphological Characterization

The macroscopic and microscopic morphology of the raw corn straw in a vertical section and a cross section are shown in Figure 1. Although both of them exhibited porous structures with pore sizes of approximately 100 μm, the cross-sectional morphology appeared more like a ‘honeycomb’, and there were many filamentous fibers perpendicular to the cross section of the straw, which enhanced the strength of the straw leading to the high strength and plastic properties [39]. Additionally, it was found that there were abundant pores with a pore size of about 30–40 μm stacked layer by layer inside the fiber in the vertical section, and most of them were rectangular without showing a honeycomb-like structure.

In the raw corn straw, cellulose is tightly packed with hemicellulose and lignin to form a rigid cell wall structure, which severely prevents esterification [40,41,42]. It can be seen in Appendix A, after the explosion treatment, that the internal fiber bundles were separated and crushed, and the hemicellulose was easily separated. After testing, the content of hemicellulose decreased from the original 29.8 wt% to 2.3 wt%.

#### 3.1.2. Esterification Efficiency

In order to clarify the impact factors on the esterification efficiency, an orthogonal experiment with three factors and four levels was designed (as shown in Appendix A). Correspondingly, through range analysis, variance analysis, and significance analysis of the test results, the primary and secondary factors affecting the efficiency and the best test scheme were determined as shown in Table 1. The results showed that the main and secondary factors affecting the degree of substitution were C > A > B, and the sample of the ECS_−4.5_, which was prepared under the optimum condition, was determined. Under a significant level of 0.05, the dosage of VA showed a significant difference.

#### 3.1.3. The Chemical Structure Characterization

As shown in Figure 2a, the FTIR spectra of the corn straw, SEC, and ECS_−x_ were demonstrated. The peaks located at 1427 cm^−1^, 1470 cm^−1^, and 1514 cm^−1^ were all attributed to the aromatic ring vibration of lignin. The wide absorption peak near 3430 cm^−1^ was assigned to the stretching vibration of -OH in the straw [43,44]. For the raw corn straw, the peak located at 1740 cm^−1^ was mainly due to the C=O stretching vibration of the hemicellulose. After the explosion treatment, the peak intensity of SEC at 1740 cm^−1^ was significantly reduced, which indicates that a large amount of hemicellulose was removed. The result was in consistent with the previous results. As for the esterified products, the peak located at 1740 cm^−1^ was assigned to the stretching vibration of the C=O in acetyl groups. Noticeably, the greater the DS, the more intensive the peak at 1740 cm^−1^, indicating that the content of C=O increased. Moreover, the esterification of cellulose or lignin with VA was studied, and the FTIR spectra (Appendix A) showed that both of them could react with VA. The FTIR spectra of all the ECS_−x_ samples are shown in Appendix A.

As shown in Figure 2b, for the HCCP, the peaks located at 1182 cm^−1^ and 873 cm^−1^ were attributed to the stretching vibration of P-N and P=N in the phosphazene ring, respectively, and the peaks located at 594 cm^−1^ and 503 cm^−1^ were both assigned to the stretching vibration of the P-Cl [45,46]. For the flame retardant (Lig-HCCP), the peak located at 960 cm^−1^ was attributed to the stretching vibration of P-O-C [46] manifesting the successful synthesis of Lig-HCCP.

Additionally, the XRD diffractogram of the SEC and ECS_−x_ are shown in Figure 3. Cellulose contained both crystalline and amorphous phases which were connected by intramolecular and intermolecular hydrogen bonds [47,48,49,50]. It can be seen that after the explosion treatment, the crystal structure of corn straw hardly changed; the curves of the corn straw and SEC contain 4 distinct and identical diffraction peaks located at 2θ = 15.1°, 16.5°, 22.6°, and 34.8° corresponding to the (101), (10$\bar{1}$), (002), and (040) crystal planes of the cellulose I crystal type. However, this also illustrated the changes in crystallinity. After the calculation (Segal method), the crystallinity of corn straw was 59.2%, while the crystallinity was reduced to 36.9% after the explosion treatment, which might improve the efficiency of the esterification. In the meantime, after the esterification, the crystal structure was destroyed. The crystallinity was decreased with the increase in the DS due to the introduction of acetyl groups, which hindered the formation of the crystalline structure.

#### 3.1.4. Thermal Properties

The thermal stability of the SEC and ECS_−4.5_ was also investigated. As shown in Figure 4a,b, the esterification had little impact on the decomposition. In addition, the Tg of the samples were obtained by DSC curves. As shown in Figure 4c, there was no glass transition found in SEC, but the glass transition temperature of ECS_−4.5_ was 186 °C. It can be seen that the introduction of the acetyl groups not only destroyed the hydrogen bond in the SEC, but also allowed the movement of the molecules more freely.

### 3.2. Morphology and Properties of the as-Prepared Corn Straw Bio-Foams

#### 3.2.1. The Morphology

The morphology of the as-prepared bio-foam was observed by SEM, and the difference between bio-foams prepared by SEC and ECS_−4.5_ was compared. As shown in Figure 5, it was easy to observe that both exhibited a porous microstructure and the pores were much smaller than the original corn straw. However, the pores in the SEC foam were larger and the density was lower when compared with the ECS_−4.5_ foam, which might be due to the strong intra-molecular hydrogen bonds in cellulose making it chemically inert and hindering the cross-linking reaction during the sol–gel process [51]. The results show that the bio-foam prepared from ECS_−4.5_ had a higher specific surface area compared with that from SEC.

#### 3.2.2. Mechanical Properties

The mechanical properties of the as-prepared bio-foam materials were evaluated by a single 40% strain compression as shown in Figure 6.

As shown in Figure 6, the as-engineered bio-foam material from the steam-exploded corn straw (SEC) had a maximum compression strength of 18.3 kPa at 40% stress. However, the compression strength for the ECS_−4.5_ bio-foam, which was derived from the esterification-modified SEC was detected to be 7.9 kPa. The bio-foam material fabricated from the ECS_−4.5_ showed more flexibility. The structure of the SEC-derived bio-foam might be slightly damaged at 40% stress and showed no resilience. Figure 6b,c respectively represent the initial state of ECS_−4.5_ foam and the recovery state after 40% strain. The ECS_−4.5_ foam shows good resilience, with an elastic recovery rate of 83% and an elastic modulus of 20 kPa. As for the Lig-HCCP-ECS_−4.5_ bio-foam material, the introduction of flame retardants did not significantly affect the porous framework, which still showed good resilience performance.

#### 3.2.3. Thermal Insulation

To further investigate the thermal insulation of the as-developed bio-foam materials, infrared thermal imaging tests were conducted as shown in Figure 7a,b.

After being heated at 101 °C for 1 h, the temperature of the upper surface point only increased to 45 °C from the initial 22.5 °C. Theoretically, the heat conduction here was mainly achieved through gas convection and the framework of the solid. The smaller the holes in the bio-foam, the greater the barrier to heat transfer [52]. Based on the previous test results, the ECS_−4.5_ and Lig-HCCP-ECS_−4.5_ foam might have smaller pores rendering better heat insulation performance.

To further illustrate the essential process of heat conduction, the thermal insulation of the samples was indicated in terms of the simulation analysis method with finite element procedures. As shown in Figure 7c,d, the parameters of physical properties (as shown in Appendix A) were used in simulation calculation, and the results are consistent with the actual situation. The Lig-HCCP-SEC_−4.5_ foam had excellent thermal insulation performance at 100 °C which resulted from the porous and dense structure of the material.

#### 3.2.4. Flame Retardancy

The limiting oxygen index (LOI) of the bio-foam materials before and after modification with the Lig-HCCP were detected. It was found that the LOI values of the steam-exploded corn straw (SEC)-derived bio-foam and the esterified steam-exploded corn straw (ECS_−4.5_) were almost the same, 23.5% and 23.7%, accordingly. After modification with the flame retardant (Lig-HCCP), the LOI of the Lig-HCCP-ECS_−4.5_ was improved to 27.2%, which was due to the flame retardant material. Combined with the previous analysis, it could be inferred that the existence of N and P in the Lig-HCCP-ECS_−4.5_ foam could be decomposed during combustion and released gas that inhibited the combustion [53]. At the same time, the lignin intrinsic flame retardance and its polyphenol structural characteristics also contributed to heat transfer.

## 4. Conclusions

In summary, corn straw was adopted as the starting material to obtain bio-foam materials with a facile dissolution–regeneration method in this work. The steam-explosion pretreatment of the pristine corn straw was conducted to obtain the fibrillated cellulose reach in lignin, which was then esterified in situ to improve the cross-linking reaction efficiency. Furthermore, the as-prepared bio-foam materials were modified with the lignin-derived flame retardant Lig-HCCP. The results show that this esterification method can successfully modify the hydroxyl groups in both cellulose and lignin, which can efficiently improve the interaction between lignin and cellulose to result in the formation of a cross-linking network. The SEC bio-foam had a maximum compression strength of 18.3 kPa at 40% stress. Although the ECS_−4.5_ bio-foam had a compressive strength of 7.9 kPa, it showed an elastic recovery ratio of 83% and an elastic modulus of 20 kPa. The as-prepared bio-foam materials also presented quite good heat insulation and flame retardance. In the future, this corn-straw-based bio-foam material could be potentially used in the production of packaging materials, thus creating alternatives for petroleum-based synthetic polymers, for example, polyurethane foam.

## Figures and Tables

**Figure 1 polymers-15-02222-f001:**
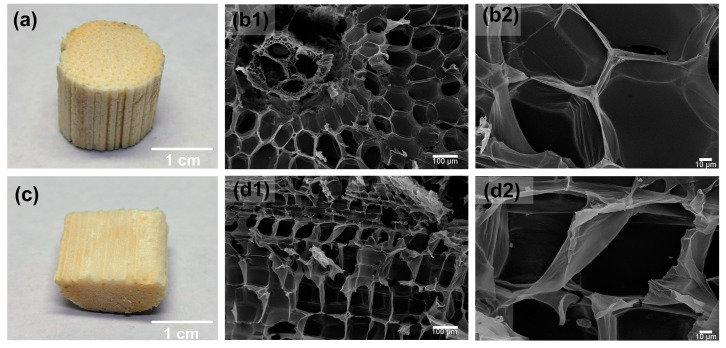
(**a**) The digital photo of raw corn straw in a cross section, (**b1**,**b2**) SEM images of raw corn straw in a cross section, (**c**) digital photo of raw corn straw in a vertical section, and (**d1**,**d2**) SEM images of raw corn straw in a vertical section.

**Figure 2 polymers-15-02222-f002:**
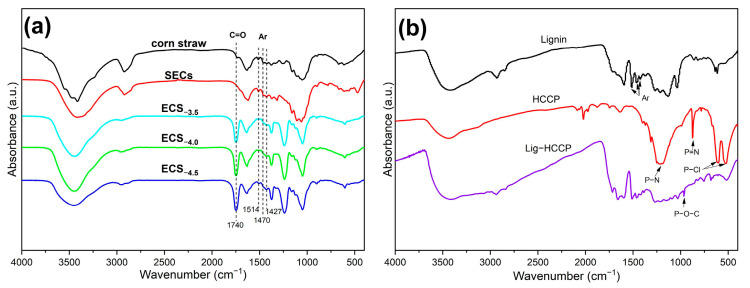
(**a**) FTIR spectra of raw corn straw, steam-exploded corn straw (SEC), and esterified steam-exploded corn straw with different degrees of substitution (ECS_−x_); (**b**) FTIR spectra of lignin, HCCP, and flame-retardant lignin (Lig−HCCP).

**Figure 3 polymers-15-02222-f003:**
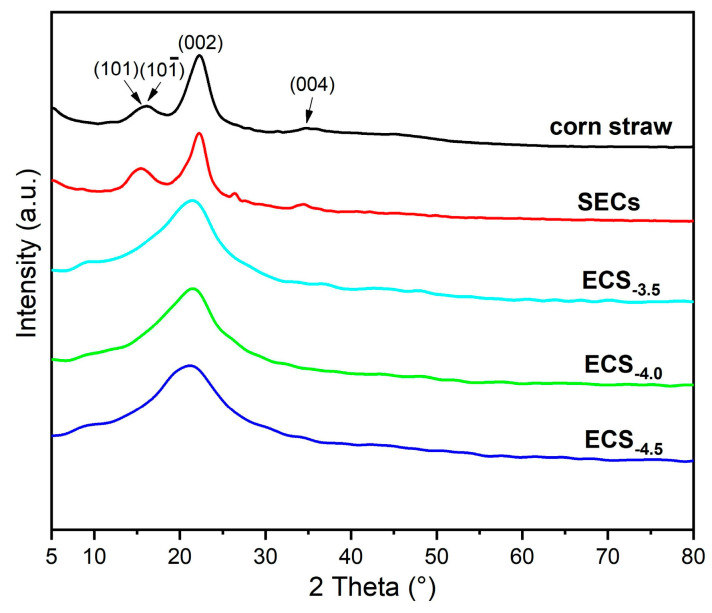
XRD diffractogram of raw corn straw, steam-exploded corn straw (SEC), and esterified steam-exploded corn straw with different degrees of substitution (ECS_−x_).

**Figure 4 polymers-15-02222-f004:**
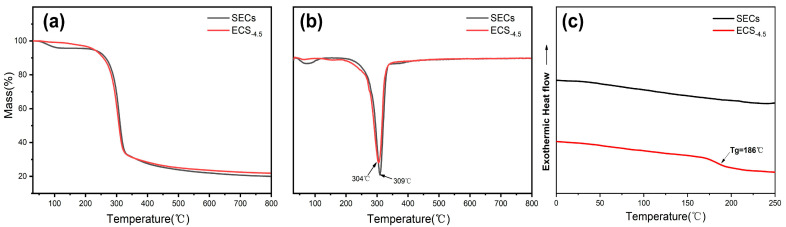
The TG (**a**), DTG (**b**), and DSC (**c**) curves of the steam-exploded corn straw (SEC) and esterified SEC (ECS-4.5).

**Figure 5 polymers-15-02222-f005:**
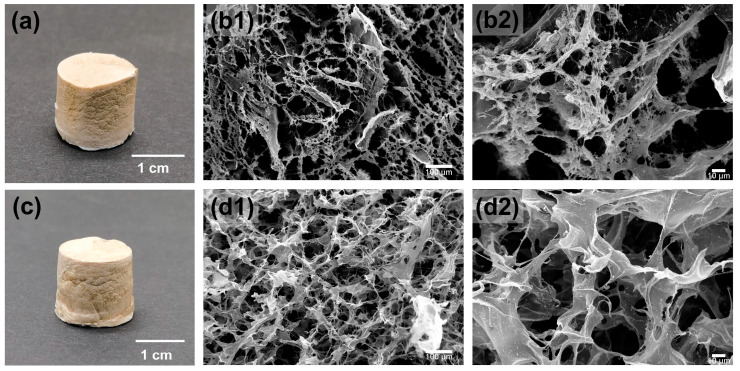
(**a**) Digital photos of the SEC bio-foam, (**b**) SEM images of the SEC bio-foam, (**b1**) was magnified 200 times, (**b2**) was magnified 1000 times (**c**) digital photos of the ECS-4.5 bio-foam, and (**d**) SEM images of the ECS-4.5 bio-foam, (**d1**) was magnified 200 times, (**d2**) was magnified 1000 times.

**Figure 6 polymers-15-02222-f006:**
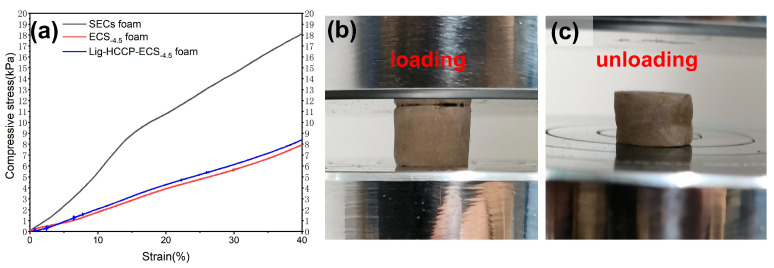
(**a**) The stress–strain curves of the SEC, ECS_−4.5_, and Lig-HCCP-ECS_−4.5_ foams; (**b**) digital photos of stress loading of the ECS_−4.5_ foam; and (**c**) digital photos of stress unloading of the ECS_−4.5_ foam.

**Figure 7 polymers-15-02222-f007:**
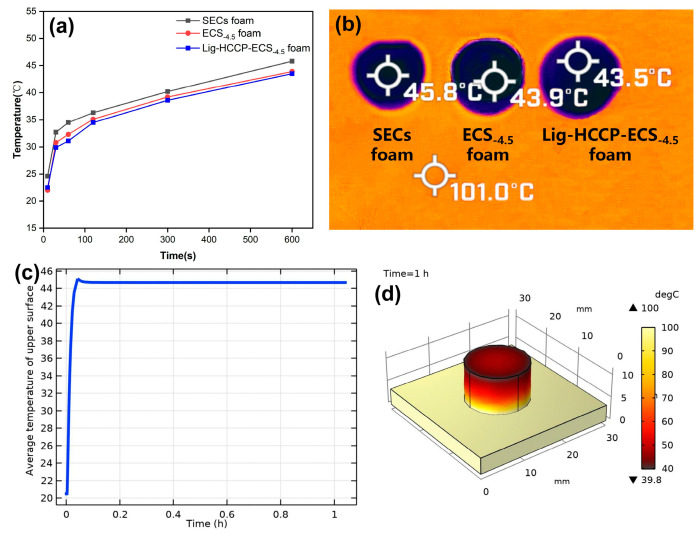
(**a**) Time-dependent curve of upper surface temperature of bio-foams on 101 °C steel plate; (**b**) upper surface temperature of bio-foams at 10 min; (**c**) thermal insulation simulation curve of Lig-HCCP-ECS_−4.5_ foam; and (**d**) thermal insulation simulation color diagram of Lig-HCCP-ECS_−4.5_ foam at 1 h time duration.

**Table 1 polymers-15-02222-t001:** Variance analysis of all factors.

	Temperature (°C)	Time (min)	Amount of VA (mol)
k1 *	4.42	4.29	3.80
k2 *	4.23	4.15	4.29
k3 *	4.06	4.49	4.67
k4 *	4.59	4.37	4.54
SS *	0.313	0.284	1.184
Mean square	0.105	0.094	0.395
Optimal level	A4	B3	C3
Influence effect	C (Amount of VA) > A (Temperature) > B (Time)		

* k1–k4: the average DS values of the four levels corresponding to each reaction factor; SS: the sum of squares of deviations.

## Data Availability

Data presented in this study are available on request from the corresponding author.

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
