# Peer review of "Preparation of Bio-Foam Material from Steam-Exploded Corn Straw by In Situ Esterification Modification"

_polymers, 2023, doi:10.3390/polym15092222_

Round 1

Reviewer 1 Report

This is a very interesting subject but not ready for publication yet. All figures, graphics and images must be redesigned. It was very hard to follow and understand the discussion referring to the actual table and/or figures presented. Graphics should be separated from images and legends give more description for the repreentations. Otherwise iteresting research loses the proper impact it should have. I recommend the authors to properly present their vauable data and disscuss it accordingly in an logic sequence.

Reviewer 2 Report

I am in favor of publishing this manuscript containing interesting data and relevant analysis. I would suggest the authors to consider following recommendations carefully and resubmit their manuscript after through revision:

1. “… which has been recognized as the most seriously trouble facing human being in this century”- the meaning of this line is not clear. Please rewrite.

2. Please correct the sentence: “With the hexachlorocyclotriphosphazene (HCCP) functionalized dealkali lignin as flame retardant, the as prepared bio-foam was modified.”

3. There are many of these errors throughout the manuscript, please correct: “Concretely, a three factor and four levels orthogonal experiment was designed and conducted as shown as Error! Reference source not found.”

4. “Fourier transform infrared spectrometer (FTIR, Bruker Tensor â…¡, Germany) was used…” – Please mention the number and rate of scans.

5. Fig. 1 a and d: Please add scale bar.

6. Please rearrange Fig. 2 (there are duplicates) and corresponding texts.

7. Table 1: Please include the units

8. “As shown in Error! Reference source not found.a, the FTIR spectra of the corn straw, SECs and ECS-x was demonstrated.” – Spectrum of raw corn straw is missing in Fig. 3a. Please include.

9. XRD data/Fig 4: Please discuss the peak assignments in the text.

10. “It could be seen that the crystallinity was reduced from 59.2% to 36.9% after explosion treatment, which might improve the efficiency of the esterification.” – This is not clear at all. How did the authors calculate the crystallinity? Please provide with the deconvolution of the spectra in the supplementary information.

11. Fig. 6 a and c: Please add scale bar.

12. Section 3.2.2: Did the authors follow any standard procedure (such as ASTM) for this characterization? Please include.

13. The English language of this manuscript needs improvement. I have mentioned some corrections in my comments, however, there are many more in the manuscript. Because of this, it is hard to understand the true meaning of many sentences. I am recommending a proofread of the manuscript thoroughly using a language editing service.

Round 2

Reviewer 1 Report

line 49 reads ...biopolymer in the earth, which is of mechanical property...

           should read....  biopolymer on earth, which is interesting for its mechanical properties and ...

Line 64 reads strong

should read strength

line 292  reads Error! Reference  source noy found

please correct this issue with the source or eraseing the source of error

Line 309-318 only fig 6a is discussed. Please discuss fig 6b and c or modify the firgure to eliminate useless images.

line 355 reads "...metrics..."  should read "...method..."
